# Age-related changes in circadian regulation of the human plasma lipidome

Shadab A. Rahman [1,2], Rose M. Gathungu [1,2,4], Vasant R. Marur [1,2,5], Melissa A. St. Hilaire [1,2,6], Karine Scheuermaier [1,2,7], Marina Belenky [1,2], Jackson S. Struble[1,2], Charles A. Czeisler[1,2], Steven W. Lockley [1,2], Elizabeth B. Klerman [1,2,3], Jeanne F. Duffy [1,2] & Bruce S. Kristal [1,2✉]

Aging alters the amplitude and phase of centrally regulated circadian rhythms. Here we evaluate whether peripheral circadian rhythmicity in the plasma lipidome is altered by aging through retrospective lipidomics analysis on plasma samples collected in 24 healthy individuals (9 females; mean ± SD age: 40.9 ± 18.2 years) including 12 younger (4 females, 23.5 ± 3.9 years) and 12 middle-aged older, (5 females, 58.3 ± 4.2 years) individuals every 3 h throughout a 27-h constant routine (CR) protocol, which allows separating evoked changes from endogenously generated oscillations in physiology. Cosinor regression shows circadian rhythmicity in 25% of lipids in both groups. On average, the older group has a ~14% lower amplitude and a ~2.1 h earlier acrophase of the lipid circadian rhythms (both, $p \leq 0.001$). Additionally, more rhythmic circadian lipids have a significant linear component in addition to the sinusoidal across the 27-h CR in the older group (44/56) compared to the younger group (18/58, $p < 0.0001$). Results from individual-level data are consistent with group-average results. Results indicate that prevalence of endogenous circadian rhythms of the human plasma lipidome is preserved with healthy aging into middle-age, but significant changes in rhythmicity include a reduction in amplitude, earlier acrophase, and an altered temporal relationship between central and lipid rhythms.

[1] Division of Sleep and Circadian Disorders, Departments of Medicine and Neurology, Brigham and Women's Hospital, 221 Longwood Ave, Boston, MA 02115, USA. [2] Division of Sleep Medicine, Harvard Medical School, Boston, MA 02115, USA. [3] Department of Neurology, Massachusetts General Hospital, Boston, MA 02114, USA. [4]Present address: Enara Bio, The Magdalen Centre, Oxford Science Park, 1 Robert Robinson Avenue, Oxford OX4 4GA, UK. [5]Present address: Quantitative Biosciences, Merck & Co., Inc, 320 Bent St, Cambridge, MA 02141, USA. [6]Present address: Department of Computer and Data Sciences, School of Science and Engineering, Merrimack College, 315 Turnpike Street, North Andover, MA 01845, USA. [7]Present address: Brain Function Research Group, School of Physiology, Faculty of Health Sciences, University of the Witwatersrand, 7 York Road, Parktown, 2193 Johannesburg, South Africa. ✉email: bkristal@bwh.harvard.edu

The regulation of circadian rhythms, which are near-ubiquitous in physiological systems, is altered with healthy aging. Comparing the rhythmic characteristics of *centrally-controlled* circadian markers in humans, including melatonin, cortisol, and core body temperature, between healthy young adults (~age 20–30 years) and middle-aged and older adults (>age 60 years), for example, shows age-associated attenuation in amplitude of circadian rhythms reported in some[1–5] but not all[4] studies and advances in circadian phase[2,3,6–13]. The phase angle between endogenous circadian phase and sleep is also altered with healthy aging, indicating that the earlier circadian phase in older individuals is not solely due to maintaining earlier wake and bedtimes. It is currently unknown, however, whether aging impacts only the circadian regulation of the *central* circadian pacemaker as assessed using markers such as melatonin and body temperature only, or whether the changes are also evident system-wide including *peripheral* oscillators and processes regulated by peripheral oscillators, such as the human transcriptome, metabolome, or lipidome. Robust circadian regulation of the human transcriptome (white blood cells), metabolome (plasma), and lipidome (plasma)[14–18] has been reported in young individuals, and 24-h daily rhythms in the human proteome have also been observed[19]. Comparing the circadian rhythm characteristics of such *peripheral* omics outcomes between younger and older individuals can provide insight into whether aging has a system-wide impact on circadian regulation.

Omics based approaches applied to blood samples from younger (<40 years) individuals have shown circadian rhythms are present in the expression of mRNA transcripts in peripheral blood mononuclear cells and metabolite levels in plasma, and the presence of diurnal rhythms in plasma protein expression. Amongst these omics-outcomes, the number of observed circadian rhythms are highest in plasma lipids, at least in younger individuals. Whether plasma lipids are also rhythmic in older individuals is unknown. Using the Constant Routine (CR) protocol, which is the gold-standard approach to minimize the impact of environmental and behavioral masking on endogenously driven circadian rhythmicity[20], ~6–9% of the human transcriptome[15,16] and ~15% of the human plasma metabolome[18] has been shown to be under endogenous circadian regulation in younger individuals. Notably, ~80% of the metabolites that exhibit a circadian rhythmicity are lipid metabolites, indicating that circadian control of lipid metabolism is more prevalent than in any other class of plasma metabolites[18]. Chua et al.[14] reported that ~13% of lipid species ($n = 263$) identified from targeted lipidomics were under circadian regulation in 20 healthy young adult men studied under CR conditions. These observations of widespread circadian regulation of lipids in humans are consistent with previous reports of circadian core-clock genes directly regulating lipid metabolism in animal models (reviewed in[21]).

Whether the age-related changes, including changes in amplitude and phase, typically observed with markers of the central pacemaker are also observed with circadian rhythmicity of peripheral circadian metabolic processes such as lipid regulation are unknown. The potential importance of investigating aging-related changes in peripheral circadian metabolic processes is supported by the growing recognition of linkages between circadian disruption and metabolic dysfunction and further underscored by the growing prevalence of metabolic disease in middle-aged and older populations. Therefore, the goal of the current study was to evaluate the effects of healthy aging on the prevalence and circadian characteristics (amplitude, phase) of the human plasma lipidome. The goal was addressed by comparing Cosinor-defined circadian profiles of lipid features between healthy young and older adults (24 individuals [9 females] in total; 12 individuals per group, 4 and 5 females in the younger and older age groups, respectively) studied independently but under equivalent protocols for at least 27 h under a CR protocol. This protocol included continuous wakefulness in a constant semi-recumbent posture, low activity level, controlled ambient temperature and dim lighting, and identical hourly meals. All participants completed the CR protocol after maintaining ~3 weeks of stable sleep-wake schedules that included sleep and wake at the same clock time during the entire interval. Blood samples collected every 3 h throughout the CR were assayed for semi-targeted plasma lipidomics.

A multistep approach was used to examine the effects of healthy aging on the prevalence and circadian characteristics of the human plasma lipidome: the prevalence (i.e., percentage of lipid metabolites which display significant rhythms) and rhythmic characteristics (phase and amplitude) of lipids were first evaluated within each age group, followed by a comparison of these endpoints between age groups. Prior to comparing circadian characteristics of lipidomics rhythms between the younger and older groups, we characterized the rhythms in each of the age groups separately and then identified the lipid species that were common between the age groups to conduct comparative analyses. Additionally, prevalence of circadian rhythms and rhythmic characteristics were evaluated at the group and individual level data by applying cosinor regression to group-averaged lipid profiles or individual-level lipid profiles.

## Results

### Sleep timing and central circadian characteristics of older and younger adults

Sleep/wake timing was earlier in older participants from Scheuermaier et al.[22], compared to younger participants from Rahman et al.[23], who were studied in equivalent protocols (Supplementary Fig. 1). Endogenous circadian phase of the melatonin rhythm was earlier in the older group than in the younger group, but the difference in relative alignment between sleep and circadian phase of the melatonin rhythm was not different between the age groups. The amplitude of the melatonin rhythm was not different between the older and younger individuals. Individual demographic details including sex, age, wake time, bedtime and endogenous central circadian phase are presented in Supplementary Table 1. The average ($\pm$SD) bedtime of the self-selected 8-h sleep episode maintained for at least 1 week immediately prior to the inpatient studies was significantly earlier in the older age group (22:29 h $\pm$ 1:00, range 21:01–23:58 h) compared to the younger group (23:54 h $\pm$ 1:12, range 21:44–1:54 h) (Watson and Williams[24] test for circular data, $p = 0.04$) (Fig. 1a). Because wake times were 16 h before bedtimes for both groups per protocol, average wake times were also earlier in the older group compared to the younger group. Therefore, by definition, and per protocol the same relationship is true between bedtime and waketime for all analysis. The average phase of the endogenous central circadian pacemaker, assessed by the dim light melatonin onset using the 25% threshold ($DLMO_{25\%}$), was significantly earlier in the older age group (20:46 h $\pm$ 1:16, range 19:00–23:04 h) compared to the younger group (21:41 h $\pm$ 1:08, range 20:13–23:52 h) (Watson and Williams[24] test for circular data, $p = 0.02$, Fig. 1b). When pooling older and younger adults together, we found a significant positive correlation between bedtime and $DLMO_{25\%}$ ($r^2 = 0.66$, $p = 0.006$, Fig. 1c). Stratified by age group this correlation was significant in both the younger ($r^2 = 0.62$, $p = 0.021$, Fig. 1c) and older age ($r^2 = 0.52$, $p < 0.0001$, Fig. 1c) groups. There was no difference in the median amplitude of melatonin between the younger and older age groups ($p = 0.62$, Fig. 1d). The phase angle of entrainment (i.e., difference in clock times between $DLMO_{25\%}$ and bedtime) was not significantly different between the younger ($DLMO_{25\%}$ 1:16 h $\pm$ 0:47 before bedtime) and the older age groups ($DLMO_{25\%}$ 1:01 h $\pm$ 0:50 before bedtime; Wilcoxon signed rank test, $p = 0.44$, Fig. 1e).

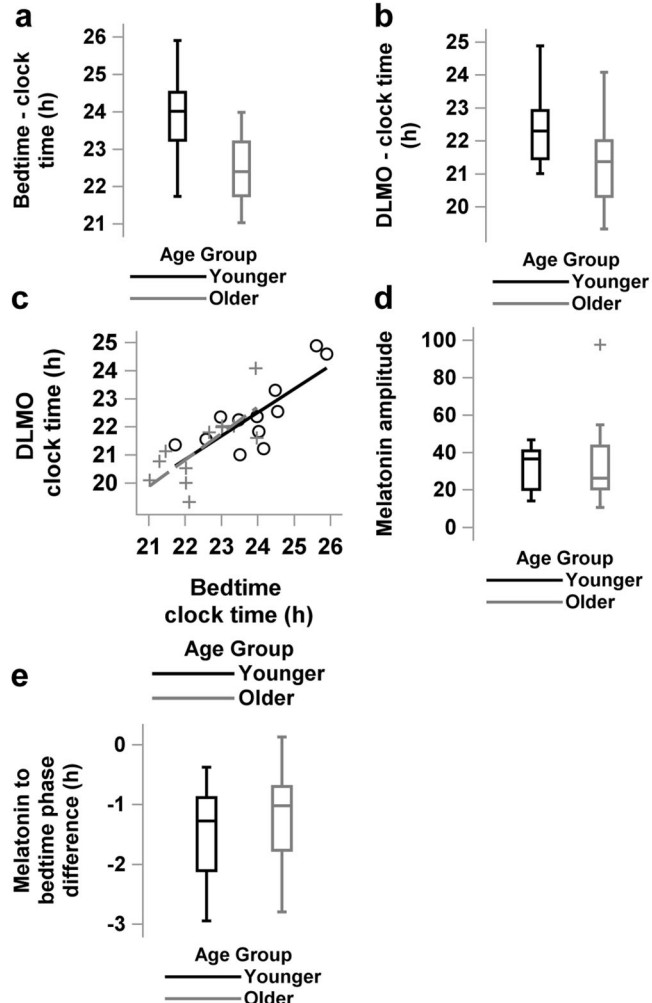

**Fig. 1 Sleep timing and central circadian timing and amplitude in younger- and older-adult age groups.** Box and whisker plots shown for younger (——, in black) and older (——, in gray) participants in the study of clock time of bedtime (**a**), and clock time of the onset of melatonin secretion (DLMO$_{25\%}$, **b**). Scatter plot of habitual bedtime and central circadian phase assessed by dim light melatonin onset (DLMO$_{25\%}$) in younger (○, black unfilled circle) and older (+, gray plus) individuals (**c**). Lines show linear regression by age group. Box and whisker plots for melatonin amplitude (**d**), and phase angle difference in hours between DLMO$_{25\%}$ and habitual bedtime (**e**) for the younger (n = 12) and older (n = 12) age groups in the study. Box and whisker plots show the median, 25th and 75th percentile (box limits), the 10th and 90th percentiles (whiskers), and outlier points.

**Circadian regulation of the human lipidome in younger adults.**
Based on the results of Cosinor regression analysis with a combined 24-h sinusoidal fundamental and a linear component, approximately one-fifth of the lipidome was under endogenous circadian regulation in the younger individuals, and almost a third of these circadian lipids also had secular linear changes (e.g., effects of accumulating sleep loss, time in semi-recumbent posture, and/or repetitive meals) during the 27-h CR. We identified 380 lipid features (here defined as a specific lipid-derived mass spectrometry peak, with a given lipid assessed only in a single electrospray mode), in some cases including isomers [e.g., three triglyceride (TG) 50:2 isomers with retention times 22.8, 23.9, 24.4 min], across 13 lipid subclasses [cholesteryl esters (CE), ceramides (Cer), diglycerides (DG), free fatty acids (FFA), lysophosphatidylcholines (LPC), lysophosphatidylethanolamines (LPE), lyso-platelet activating factor

(lyso-PAF), phosphatidylcholines (PC), phosphatidylethanolamines (PE), phosphatidylglycerol (PG), phosphatidylinositol (PI), sphingomyelins (SM), and TG].

Of the 380 lipid features, 81 were circadian at the group-level, of which 24 showed combined 24-h rhythmic sinusoidal and linear changes across the CR (Supplementary Fig. 2a), and 57 showed only 24-h rhythmic sinusoidal changes (Supplementary Fig. 2b). In addition, 93 of the 299 lipid features that did not exhibit a sinusoidal circadian rhythm had significant linear changes across the 27-h CR interval (Supplementary Fig. 2c, Table 1). Across the CR interval, these linear lipid features changed levels by up to 1.44 SDs from the mean per 24 h of wakefulness (0.96 SDs per a typical 16-h wake episode; Table 1).

Circadian rhythms were detected in almost all lipid subclasses except for CEs (n = 13 lipid features), FFAs (n = 19 lipid features), and Lyso-PAF and PG (n = 1 lipid feature in both subclasses) (Fig. 2a, Table 1). The prevalence of circadian rhythms within a lipid subclass that had at least one circadian feature ranged from ~7% (TG: 15/210 lipid features) to 100% (PE and PI: 10/10 lipid features in both subclasses). The amplitude of the circadian lipid profiles, calculated as half the displacement of the sinusoidal profile fit to z-scored lipid levels, ranged between 0.30 and 0.67 standard deviations from the mean in SM and LPE subclasses, respectively, throughout the 24-h day. The prevalence estimates and circadian characteristics across each of these lipid subclasses are reported in Table 1. The majority of the circadian lipids had relatively lower plasma concentrations overnight and at the start of the CR, and the levels of these lipids were relatively higher during the waking day (i.e., the first 16 h of CR), most often peaking between 7.5 and 16.5 h after wake (Fig. 2b). The majority of the lipid subclasses had acrophases (peak timing of the fitted rhythm) ~12 h after waking, with an ~8-h range in the acrophases between lipid subclasses (Fig. 2c, Table 1).

**Circadian regulation of the human lipidome in older adults.**
Circadian regulation of the lipidome was observed in the older group at essentially equal prevalence to that observed in the younger group. We identified higher number of unique lipids and isomers in the older age group (417 lipids features). Of the 417 lipid features, 89 were circadian, of which 68 showed combined rhythmic (24-h) and linear changes across the CR (Supplementary Fig. 2d), and 21 showed only 24-h rhythmic changes (Supplementary Fig. 2e). In addition, 114 of the 328 lipid features that did not exhibit a circadian rhythm had significant linear changes across the 27-h CR interval (Table 2, Supplementary Fig. 2f). Across the CR interval, these linear lipid features changed levels by up to 1.54 SDs from the mean per 24 h of wakefulness (1.02 SDs per a typical 16-h wake episode; Table 2).

Circadian rhythms were detected in almost all lipid subclasses except for PGs (n = 4 lipid features) and SMs (n = 26 lipid features) (Fig. 3a, Table 2) in older individuals. The prevalence of circadian rhythms in lipid features ranged from ~6% (FA: 1/17 lipid features) to ~67% (LPE: 2/3 lipid features). The circadian lipids also exhibited robust oscillations with amplitudes ranging between 0.14 and 0.50 standard deviations from the mean in the FA and LPC subclasses, respectively, throughout the 24-h day. The prevalence estimates and circadian characteristics across each of these lipid subclasses are reported in Table 2. The majority of the circadian lipids had relatively lower plasma concentrations in the morning at the start of the CR and peaked overnight toward the end of the CR (Fig. 3b). Even though the plasma lipid levels peaked toward the end of the CR, the majority of the circadian lipids had an acrophase estimated by the Cosinor regression around 10 h after wake with approximately a 10-h range in acrophases between lipid subclasses (Fig. 3c, Table 2).

**Table 1 Circadian characteristics in the plasma lipidome of younger healthy individuals.**

| Lipid | Total | Circadian | Linear | Amplitude | | | Acrophase | | Slope (Δ/24-h) | | |
|---|---|---|---|---|---|---|---|---|---|---|---|
| | | | | Median | 25th Pctl | 75th Pctl | Circ. Mean | Circ. SD. | Median | 25th Pctl | 75th Pctl |
| CE | 13 | 0 | 1 | – | | | | | 1.24 | 1.24 | 1.24 |
| Cer | 21 | 7 | 2 | 1.04 | 1.04 | 1.04 | 1.04 | 1.04 | 1.10 | 0.77 | 1.43 |
| DG | 11 | 2 | 3 | 0.42 | 0.35 | 0.49 | 6.21 | 0.88 | 1.12 | 1.03 | 1.17 |
| FFA | 19 | 0 | 7 | – | | | | | 0.63 | 0.60 | 0.65 |
| LPC | 17 | 5 | 0 | 0.62 | 0.51 | 0.64 | 12.26 | 0.39 | | | |
| LPE | 3 | 2 | 1 | 0.67 | 0.65 | 0.68 | 11.84 | 0.08 | 1.16 | 1.16 | 1.16 |
| Lyso-PAF | 1 | 0 | 0 | – | | | | | | | |
| PC | 41 | 26 | 5 | 0.40 | 0.37 | 0.47 | 11.33 | 1.61 | 1.21 | 1.10 | 1.32 |
| PE | 10 | 10 | 0 | 0.43 | 0.41 | 0.48 | 12.37 | 0.26 | – | | |
| PG | 1 | 0 | 0 | – | | | | | | | |
| PI | 10 | 10 | 0 | 0.49 | 0.44 | 0.52 | 13.80 | 2.48 | – | | |
| SM | 23 | 4 | 0 | 0.30 | 0.27 | 0.33 | 9.34 | 0.39 | – | | |
| TG | 210 | 15 | 74 | 0.37 | 0.31 | 0.42 | 2.48 | 4.04 | 1.24 | 1.07 | 1.49 |

*CE* cholesteryl esters, *Cer* ceramides, *DG* diglycerides, *FFA* free fatty acids, *LPC* lysophosphatidylcholines, *LPE* lysophosphatidylethanolamines, *Lyso-PAF* lyso-platelet activating factor, *PC* phosphatidylcholines, *PE* phosphatidylethanolamines, *PG* phosphatidylglycerol, *PI* phosphatidylinositol, *SM* sphingomyelins, *TG* Triglyceride, *Pctl* Percentile, *Circ* Circular, *SD* Standard Deviation, Δ Change in Z-score normalized levels.

**Aging-associated increase in linear change with wakefulness among circadian-regulated lipids**. As noted above, individuals in both the younger and older groups had similar proportions of lipids that displayed circadian rhythmicity, but the proportion of lipids that did or did not have combined linear changes differed between the age groups. This suggests the possibility that although the overall circadian regulation of the lipidome is preserved in both age groups, among older individuals, circadian regulation of the lipidome is differentially susceptible to perturbations induced by external factors such as sleep loss or frequent meals around the clock during CR compared with younger individuals.

Specifically, there were significantly more circadian-regulated lipids containing a linear term with a combined 24-h harmonic in the older ($n = 68/89$, 76%) than in the younger age group ($n = 24/81$, 30%, Fig. 4b), and conversely, there were significantly more circadian-regulated lipids that showed only 24-h harmonic changes without a concurrent linear change over the duration of wakefulness in the younger ($n = 57/81$, 70%) than in the older participants ($n = 21/89$, 24%, χ2, $p < 0.0001$, Fig. 4b). The proportion of lipids that changed only linearly without any significant 24-h harmonic changes over the 27-h CR in the older ($n = 114/328$, 35%) compared to the younger age group ($n = 93/299$, 31%) was not significantly different, however (χ2, $p = 0.29$, Fig. 4b).

**Circadian characteristics of the plasma lipidome with healthy aging**. Circadian rhythmicity was proportionally similar between the younger and older groups, but the amplitude and timing of the circadian rhythms displayed age-related changes. Of the 235 lipids that were present in both age groups, 89 unique lipids displayed endogenous circadian rhythmicity in at least one of the two groups (Fig. 4a, Supplementary Fig. 3). The proportion of lipids that were circadian regulated did not differ statistically between the younger (56/235) and older (58/235) groups (χ2, $p = 0.86$, Fig. 4b, Supplementary Fig. 4).

As a first pass, we compared the average amplitude of the 89 lipids that were circadian-regulated in either group, which showed a ~ 16% reduction in average amplitude in the older group (mean ± SD: 0.38 ± 0.10, $n_{lipids} = 56/89$) compared to the younger group (0.44 ± 0.11, $n_{lipids} = 58/89$, GLM, $p = 0.001$, Fig. 4c). This difference remained significant (GLM, $p = 0.001$) after adjusting for whether the lipid rhythm included a combined 24-h harmonic and linear component or a 24-h harmonic only

($p = 0.01$, Supplementary Fig. 5) and for lipid subclasses ($p < 0.0001$, Supplementary Fig. 6).

Comparing the acrophase of the 89 circadian lipids showed that, on average, the peak of the underlying circadian rhythm occurred earlier in the older group compared to the younger group. This earlier phase of the rhythms in the older group, compared to the younger group, occurred despite more lipids reaching maximal plasma concentrations in this group at the end of the CR. Estimating the acrophase of the sinusoidal rhythm after adjusting for linear changes in lipid levels associated with secular effects during CR (e.g., sleep deprivation, frequent meals around the clock; and captured by the linear component of the Cosinor model) showed that the mean acrophase of circadian lipids occurred significantly earlier in the older (9.3 h ± 1.7 h after wake) compared to the younger group (11.4 ± 1.1 h after wake, Watson and Williams[24] test for circular data, $p < 0.0001$, Fig. 4e). As one potential reason for the reduction in average amplitude across multiple lipids in the older group may have been increased variance in the acrophase of individual lipids, we compared the circular variance in acrophase of the circadian lipids between the groups and did not find a statistically significant difference between the older (1.7 h) and younger group (1.1 h, Wilcoxon, $p = 0.07$, Fig. 4f).

To test whether the observed differences in amplitude and acrophase between the two age groups were due to comparing different lipid species between the age groups, we further compared the circadian characteristics between the two age groups but limited the analyses to only those lipid species that displayed circadian rhythmicity in both groups. Of the 94 lipid species that were circadian in at least one group, 25 were common to both age groups ($p < 0.005$ by binomial probability given 25/56 matches, Fig. 5a). The mean acrophase of the 25 common circadian lipids occurred significantly earlier in the older (9.6 h ± 1.6 h after wake) compared to the younger participants (11.9 ± 0.3 h after wake, Watson and Williams test for circular data, $p < 0.0001$, Fig. 5c), and the variance in acrophase was not different between the groups (Wilcoxon, $p = 0.57$, Fig. 5d). These data are functionally equivalent to that seen in the entire lipid series.

Circadian amplitude was, on average, ~29% higher in the younger (0.49 ± 0.11) than in the older group for these 25 lipids (0.38 ± 0.10, GLM, $p = 0.001$, Fig. 5b). Additionally, within each of the lipid subclasses covering these 25 lipids, amplitude appeared to be higher in the younger group compared to the older group except

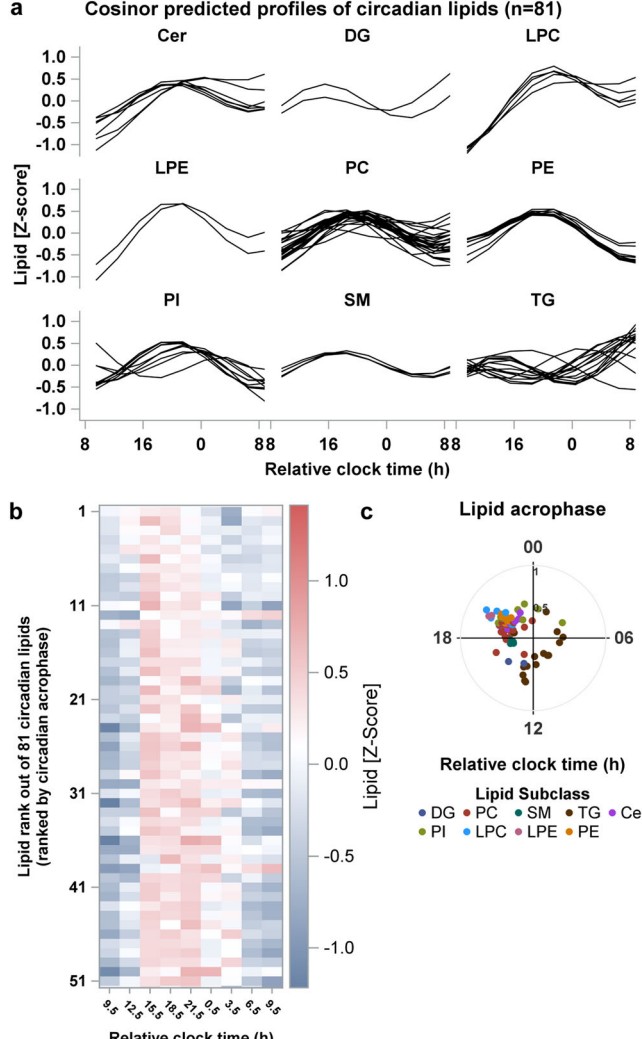

**Fig. 2 Prevalence, timing and amplitude of circadian rhythms in the plasma lipidome of younger-adults age group adults. a** Predicted time-course profiles across the 27-h constant routine are shown for each circadian lipid species in each lipid subclass (total of 81 rhythmic lipids across subclasses). **b** Heat map for Z-scored plasma concentration levels of each circadian lipid. Lipids are ranked based on the time of the acrophase of each lipid estimated from the Cosinor regression applied to group-mean data for each lipid. **c** Polar plots of acrophase of each circadian lipid. Data are plotted to a relative clock time with wake time at the start of the constant routine assigned a value of 8:00 am. The radial axis represents the amplitude estimate for the lipid. Results are from $n = 12$ biologically independent samples in the younger age group. CE cholesteryl esters, Cer ceramides, DG diglycerides, FFA free fatty acids, LPC lysophosphatidylcholines, LPE lysophosphatidylethanolamines, Lyso-PAF lyso-platelet activating factor, PC phosphatidylcholines, PE phosphatidylethanolamines, PG phosphatidylglycerol, PI phosphatidylinositol, SM sphingomyelins, TG Triglyceride.

in the CER subclass. We observed a trend linking lipid subclass to the effect of aging on circadian amplitude (GLM with main effects and interaction between age-group and lipid subclass, $p = 0.06$, Supplementary Fig. 7). Due to the small sample size, however, there is limited statistical power to further investigate this. Additionally, given that the circadian amplitude is expressed at the level of z-scores, it likely underestimates the differences (i.e., the age-related loss) in amplitude, as the mean biological Coefficients of Variation (CV) are significantly (linear mixed model, $p < 0.0001$) lower in the

older compared to the younger adults (mean $0.25 \pm 0.02$ vs. $0.58 \pm 0.02$ [median $= 0.21$ vs. $0.43$], respectively). To minimize the possible confounding of temporal changes in lipid levels affecting the CV estimates, the CV estimates were calculated in those lipids that had temporally stable levels across the CR (i.e., displayed neither a linear trend nor a circadian rhythm).

**Results at the individual level mirror group-level observations.** Circadian characteristics, including estimates of prevalence, amplitude and acrophase, may differ based on whether the estimates are derived from group- or individual-level data[14,17]. We therefore re-evaluated the differences in circadian characteristics between the age groups with individual-level data. Among individuals, 182 unique lipids had a significant circadian component in at least one individual in at least one age group. *Prevalence*: Across 12 individuals in the younger group, ~20% ($n = 441$) of the lipid features were circadian out of a possible total of 2184 circadian features (i.e., if all 182 lipids that display circadian rhythmicity in at least 1 individual were circadian in all 12 individuals). In the older group, ~21% ($n = 454/2184$) of the lipid features displayed circadian rhythmicity. *Acrophase:* Most lipid species peaked during the biological day both in the younger and older groups (Fig. 6a). The mean acrophase of the circadian lipids occurred significantly later in the younger ($12.4 \pm 2.7$ h after wake) compared to the older ($10.5$ h $\pm 2.7$ h after wake) group (Watson and Williams test for circular data, $p < 0.0001$), consistent with the later phase of the group-averaged waveforms. Although the variance in acrophases was higher when calculated using individual-level data, as compared to group-level data, the variance was not significantly different between the groups (Wilcoxon, $p = 0.93$, Fig. 6b). *Amplitude*: Median amplitude, when calculated using individual-level data was significantly lower in the older group compared to the younger age group (Wilcoxon, $p < 0.0001$, Fig. 6c), consistent with reduced amplitude of the group-averaged waveforms in the older group.

**Association between circadian rhythms in the plasma lipidome and melatonin in younger and older adults.** The plasma melatonin rhythm is a reliable indirect measure of the output of the central circadian clock. Therefore, we compared the relative alignment of peripheral lipid rhythms and the plasma melatonin rhythm between younger and older individuals. We limited this analysis at the group level to the 25 circadian lipids that were present both in the younger and older groups. For each age group, alignment of each of the 25 lipids relative to the melatonin profile was estimated by calculating the difference in hours between the acrophase of the group-mean lipid profile and the melatonin phase ($DLMO_{25\%}$) averaged across individuals in that age group. On average, the phase difference between lipid circadian rhythms and melatonin was larger in the younger than in the older group ($-4.28$ h $\pm 1.65$ vs. $-1.31$ h $\pm 4.26$, respectively, paired $t$-test $p = 0.001$, Fig. 6d). Linear regression analysis of lipid to melatonin phase angle difference between younger and older individuals showed a slope of 1.00 ($p = 0.053$), suggesting that the age-related phase angle difference was equal for all 25 lipids (Fig. 6e).

Pooled across both age groups, the prevalence of circadian lipids with only a sinusoidal pattern (i.e., with no linear change throughout the CR) increased with increasing melatonin rhythm amplitude even after adjusting for age (GEE, $p < 0.0001$, Fig. 6f). When stratified by age group, the association remained significant in the older group ($p < 0.0001$) but was not significant in the younger group ($p = 0.70$). Similarly, for lipid species that had a combined sinusoidal and linear change during the CR, their prevalence also increased with increasing melatonin rhythm amplitude, pooled across both age groups and adjusted for age ($p = 0.004$). Again, when stratified by age group, this association

**Table 2 Circadian characteristics in the plasma lipidome of older healthy individuals.**

| Lipid | Total | Circadian | Linear | Amplitude | | | Acrophase | | Slope (Δ/24-h) | | |
|---|---|---|---|---|---|---|---|---|---|---|---|
| | | | | Median | 25th Pctl | 75th Pctl | Circ. Mean | Circ. SD. | Median | 25th Pctl | 75th Pctl |
| CE | 18 | 4 | 3 | 0.32 | 0.28 | 0.37 | 16.54 | 7.34 | −1.32 | −1.70 | −0.89 |
| Cer | 21 | 3 | 1 | 0.44 | 0.38 | 0.45 | 7.75 | 3.01 | 0.94 | 0.94 | 0.94 |
| DG | 12 | 7 | 1 | 0.33 | 0.28 | 0.36 | 9.36 | 3.80 | 1.29 | 1.29 | 1.29 |
| FA | 17 | 1 | 2 | 0.14 | 0.14 | 0.14 | 12.03 | — | 1.21 | 1.02 | 1.40 |
| LPC | 17 | 11 | 3 | 0.50 | 0.39 | 0.53 | 9.70 | 1.73 | 1.13 | 1.09 | 1.70 |
| LPE | 3 | 2 | 1 | 0.48 | 0.42 | 0.53 | 12.68 | 0.26 | 0.92 | 0.92 | 0.92 |
| Lyso-PAF | 1 | — | | | | | | | | | |
| PC | 93 | 24 | 43 | 0.37 | 0.30 | 0.42 | 5.19 | 2.44 | 1.16 | 0.90 | 1.48 |
| PE | 28 | 10 | 11 | 0.25 | 0.20 | 0.28 | 16.28 | 3.15 | 0.87 | 0.82 | 1.04 |
| PG | 4 | — | 2 | — | | | | | 0.64 | 0.57 | 0.70 |
| PI | 19 | 8 | 9 | 0.36 | 0.32 | 0.39 | 8.32 | 1.16 | 1.22 | 1.16 | 1.31 |
| SM | 26 | — | | | | | | | | | |
| TG | 158 | 19 | 38 | 0.43 | 0.32 | 0.46 | 11.24 | 2.80 | 1.37 | 1.10 | 1.54 |

CE: cholesteryl esters, Cer: ceramides, DG: diglycerides, FFA: free fatty acids, LPC: lysophosphatidylcholines, LPE: lysophosphatidylethanolamines, Lyso-PAF: lyso-platelet activating factor, PC: phosphatidylcholines, PE: phosphatidylethanolamines, PG: phosphatidylglycerol, PI: phosphatidylinositol, SM: sphingomyelins, TG: Triglyceride; Pctl: Percentile; Circ: Circular, SD: Standard Deviation, Δ: Change in Z-score normalized levels.

was significant in the older group ($p = 0.001$) but not in the younger group ($p = 0.52$).

## Discussion

Overall, we found that the prevalence of circadian rhythmicity in the human plasma lipidome was unchanged with age, but there was a significant advance in the phase and a significant reduction in the amplitude of circadian lipid profiles in older compared to younger adults. These results were consistent when data were examined at the group and individual level, demonstrating the robustness of the findings. Results from our study corroborate evidence that the human plasma lipidome is under endogenous circadian regulation in healthy young adults[14] and extend this evidence to show the impact of healthy aging on the circadian regulation of the human plasma lipidome.

The current report provides an initial foundational resource for the investigation of several key questions on potential linkages between age-associated weakening of circadian regulation of plasma lipidomics and age-associated metabolic dysfunction. Specifically, we provide evidence that, at least through middle-age, the relative prevalence of circadian rhythmicity in lipids is maintained; however, they show lower amplitudes, and are more affected by non-circadian influences (e.g., sleep deprivation). There is also little evidence for increased variability in these systems with increasing age. In addition, both the central clock and its peripheral outputs shift to an earlier time, potentially exposing individuals to misalignment between, for example, the timing of meals relative to endogenous central circadian phase. Moreover, the phase angle between centrally-controlled markers (i.e., melatonin) and peripheral outputs (i.e., plasma lipids) of circadian rhythmicity was altered, suggesting that the circadian network decoupling that occurs with aging may also be driving downstream metabolic dysfunction.

The prevalence data from our study show that, in general, the circadian profile as assessed by Cosinor analysis of the human plasma lipidome may not be lost with healthy aging, as ~20–21% of the lipid species were under endogenous circadian regulation in both younger and older individuals, but the data examining linear trends across sleep deprivation (during the CR) demonstrate an age-associated reduction in the ability to maintain purely sinusoidal circadian oscillations in the presence of non-circadian influences on lipid levels (e.g., sleep deprivation). Specifically, significantly more lipid circadian rhythms had combined sinusoidal and linear components, representing secular changes with wakefulness, in the older individuals compared to younger individuals. Additionally, significantly more lipid profiles changed only linearly (i.e., without a significant 24-h sinusoidal component) during the constant routine protocol in the older than in the younger age group. While our study is not designed to identify the causal source of this evoked linear component, possibilities include homeostatic responses to sustained wakefulness and/or frequent meals. Nonetheless, the results suggest that external behavioral factors may influence the apparent expression of circadian rhythm of plasma lipids more in older adults than in younger adults.

Our observations that circadian amplitude of the lipid rhythms was significantly reduced in older compared to younger individuals strengthens the evidence supporting age-associated decrements in circadian strength which may be replaced with an increased influence by external factors such as sleep loss and frequent meals. Our finding is consistent with previous reports of reduced amplitude of central circadian clock markers, including melatonin, core body temperature and cortisol rhythms in older individuals compared to younger individuals[1–5]. A reduction in melatonin amplitude with aging has not been found in all studies[4], and the lower amplitude of melatonin in the older group compared to the young group in the current study also did not reach statistical significance.

One possibility for the observed reduction in amplitude in lipid circadian rhythms may have been increased variance in the phase of the individual lipid rhythms, which when averaged together resulted in an attenuated estimate of the overall amplitude[14,25]. This, however, appears unlikely because we did not find a difference in the variance in phase between the age groups using group- or individual-level analyses. Due to relatively low statistical power to test this endpoint in this study, confirmation will require follow-up studies.

The reduction in amplitude and/or the susceptibility to external influences may be due to reduced output strength (i.e., ability to synchronize rhythms) of the central pacemaker or reduced recognition of the synchronizing signal at the downstream targets. While our current study precludes identifying whether this loss of amplitude is due to change in the output strength of the central pacemaker or other peripheral oscillators, or recognition of the synchronizing signals at the downstream targets, studies in animal models do support a role of the central pacemaker in

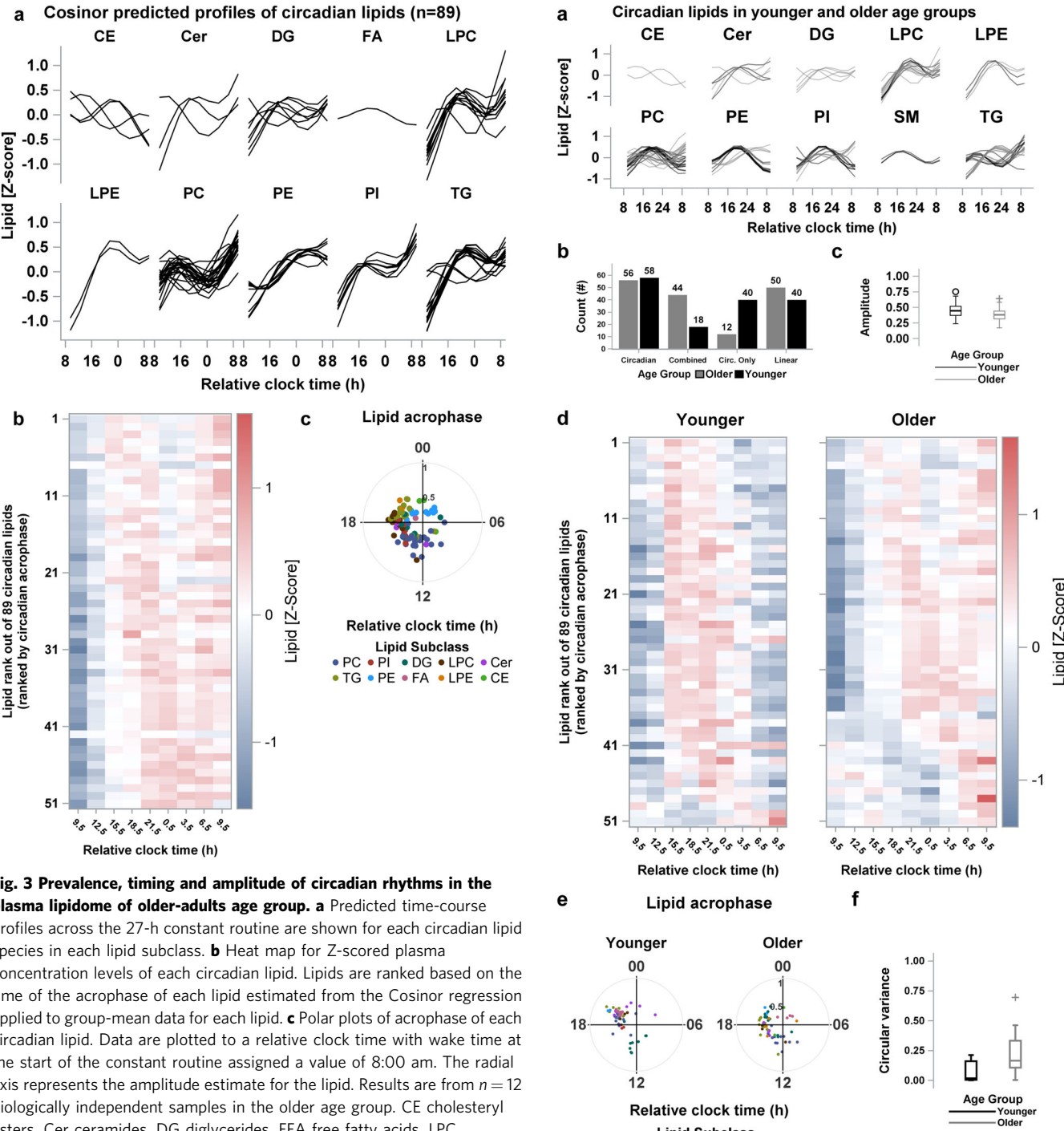

**Fig. 3 Prevalence, timing and amplitude of circadian rhythms in the plasma lipidome of older-adults age group. a** Predicted time-course profiles across the 27-h constant routine are shown for each circadian lipid species in each lipid subclass. **b** Heat map for Z-scored plasma concentration levels of each circadian lipid. Lipids are ranked based on the time of the acrophase of each lipid estimated from the Cosinor regression applied to group-mean data for each lipid. **c** Polar plots of acrophase of each circadian lipid. Data are plotted to a relative clock time with wake time at the start of the constant routine assigned a value of 8:00 am. The radial axis represents the amplitude estimate for the lipid. Results are from $n = 12$ biologically independent samples in the older age group. CE cholesteryl esters, Cer ceramides, DG diglycerides, FFA free fatty acids, LPC lysophosphatidylcholines, LPE lysophosphatidylethanolamines, Lyso-PAF lyso-platelet activating factor, PC phosphatidylcholines, PE phosphatidylethanolamines, PG phosphatidylglycerol, PI phosphatidylinositol, SM sphingomyelins, TG Triglyceride.

affecting the amplitude of peripheral rhythms. Fetal SCN transplant in aged golden hamsters increases the amplitude of rhythms in locomotor activity, temperature, drinking, and transcription of corticotropin-releasing hormone in the brain, suggesting that changes within the aging SCN contributes to the reduced amplitude of overt and peripheral rhythms[26,27]. Furthermore, in a postmortem study among older individuals (average age of 90 years), albeit older than the middle-aged individuals in the current study, vasoactive intestinal polypeptide (VIP) expression in the SCN correlated positively with the amplitude of activity

rhythms in these individuals based on actigraphy measures taken within 18 months of death, suggesting that changes within the SCN with aging in humans may contribute to observed changes in peripheral rhythms[28]. Interestingly, the amplitude of core clock gene expression rhythms in individual SCN neurons remains unchanged with aging, suggesting that core clock machinery is unaffected by aging[29]. In contrast, the amplitude of SCN neuronal firing rhythms and neurotransmitter expression including arginine vasopressin (AVP) and VIP, is reduced with aging, suggesting that output strength and the ability of the SCN to synchronize peripheral rhythms is likely diminished with aging[30,31]. This decline is consistent with our observation of a

**Fig. 4 Comparative analysis of the prevalence, timing and amplitude of circadian rhythms in the plasma lipidome of younger- and older-adults age groups. a** Predicted time-course profiles calculated from group-mean data across the 27-h constant routine are shown for each circadian lipid species in each lipid subclass for the younger (——, in black) and older (——, in gray) age groups. **b** Prevalence of circadian and linear lipid profiles during the 27-h constant routine in the younger and older age groups. Circadian profiles were further dichotomized to be "Combined" when both the sinusoidal and linear terms were significant or only circadian (Circ. only) when the sinusoidal but not the linear term was significant in the Cosinor regression. **c** The box and whisker plots for amplitude estimates for circadian lipids for the younger and older age groups. **d** Heat maps for Z-scored plasma concentration levels of each circadian lipid in the younger and older age groups. Lipids are ranked based on the time of the acrophase of each lipid estimated from the Cosinor regression applied to group-mean data for each lipid. **e** Polar plots of acrophase of each circadian lipid in the younger and older age groups. Data are plotted to a relative clock time with wake time at the start of the constant routine assigned a value of 8:00 am. The radial axis represents the amplitude estimate for the lipid. **f** The box and whisker plots of circular variance of lipid acrophases for the younger and older age groups. The box and whisker plots show the median, 25th and 75th percentile (box limits), the 10th and 90th percentiles (whiskers), and outlier points. Results are from $n = 24$ biologically independent samples in the younger ($n = 12$) and older ($n = 12$) age groups. CE cholesteryl esters, Cer ceramides, DG diglycerides, FFA free fatty acids, LPC lysophosphatidylcholines, LPE lysophosphatidylethanolamines, Lyso-PAF lyso-platelet activating factor, PC phosphatidylcholines, PE phosphatidylethanolamines, PG phosphatidylglycerol, PI phosphatidylinositol, SM sphingomyelins, TG Triglyceride.

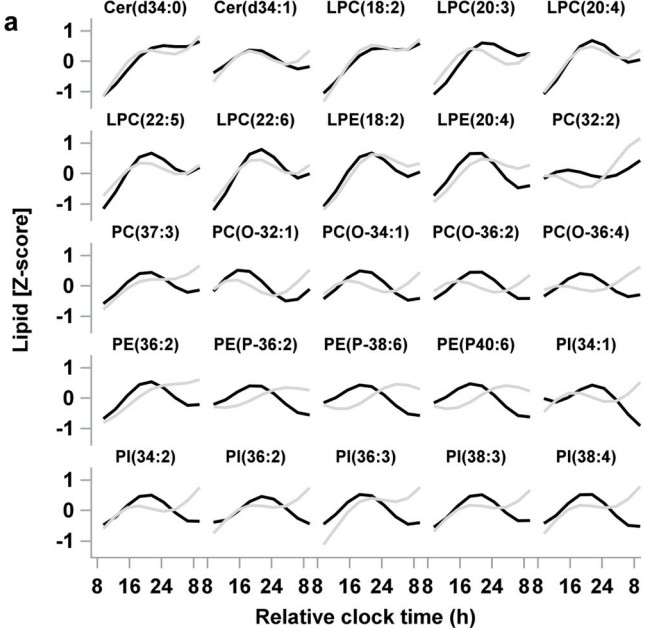

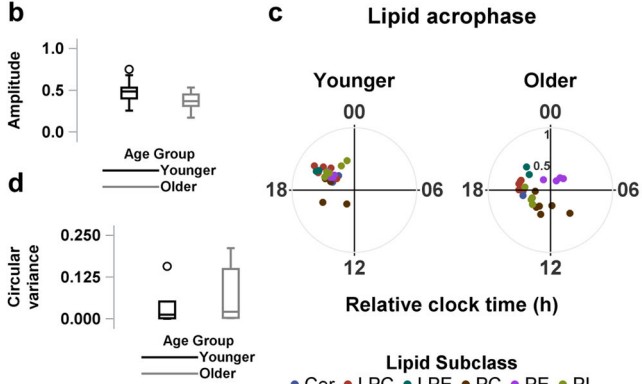

reduced amplitude of lipid circadian rhythmicity in the older age group as well as the increased proportion of circadian lipids changing linearly across the CR, which indicates greater susceptibility of the weakened endogenous circadian oscillations to external influences. We identified a significant association between the number of lipid rhythms identified as circadian and the amplitude of the melatonin rhythm. Indeed, this relationship with amplitude was significantly stronger in the older individuals.

A recent report examining 24-h rhythmicity in gene expression profiles from RNA-seq samples collected from 914 donors across 46 tissues found that rhythmicity was largely conserved across tissues in both younger ($38 \pm 9$ years) and older ($65 \pm 3$ years) groups, rhythmic gene expression was largely damped in older individuals[32]. Importantly, however, these rhythms were not assessed under constant routine conditions, which precludes conclusively determining whether these observed changes are strictly due to changes in endogenous circadian regulation with aging or the reflect changes in the impact of external behavioral and environmental influencers (e.g., sleep/wake, activity, feeding) on circadian regulation and peripheral clocks in humans. Nonetheless, these results are largely consistent with our findings that endogenous circadian regulation of peripheral rhythms in lipids is conserved across aging but significantly damped (i.e., lower amplitude) in older humans. Furthermore, both of our results are also consistent with previous reports from studies in animal models that show that aging affects peripheral clocks with phase advances and damped oscillations, which, data suggests, is due to reduced coupling strength of the SCN to peripheral tissues[33,34].

Our finding that lipid rhythms are advanced (i.e., earlier acrophase) in older individuals compared to younger individuals is consistent with previous reports of an earlier phase in central circadian output measures, i.e., melatonin, cortisol and core body temperature with aging[2,3,6–13]. Our results suggest that the earlier circadian phase observed in older people is not limited to a few

**Fig. 5 Comparative analysis of the timing and amplitude of the lipid species that were circadian in both the younger- and older-adult age groups. a** Predicted time-course profiles calculated from group-mean data across the 27-h constant routine are shown for each circadian lipid species that was circadian in both the younger (——, in black) and older (——, in gray) age groups. **b** The box and whisker plots of amplitude estimates for circadian lipids for the younger and older age groups. **c** Polar plots of acrophase of each circadian lipid in the younger and older age groups. Data are plotted to a relative clock time with wake time at the start of the constant routine assigned a value of 8:00 am. The radial axis represents the amplitude estimate for the lipid. **d** The box and whisker plots of circular variance of lipid acrophases for the younger and older age groups. The box and whisker plots show the median, 25th and 75th percentile (box limits), the 10th and 90th percentiles (whiskers), and outlier points. Results are from $n = 24$ biologically independent samples in the younger ($n = 12$) and older ($n = 12$) age groups. CE cholesteryl esters, Cer ceramides, DG diglycerides, FFA free fatty acids, LPC lysophosphatidylcholines, LPE lysophosphatidylethanolamines, Lyso-PAF lyso-platelet activating factor, PC phosphatidylcholines, PE phosphatidylethanolamines, PG phosphatidylglycerol, PI phosphatidylinositol, SM sphingomyelins, TG Triglyceride.

key centrally-controlled circadian phase markers such as melatonin and core body temperature, but is pervasive, occurring in both centrally- and peripherally-regulated circadian rhythms such as plasma lipids. The shift to an earlier phase in lipid rhythms is likely associated with the significantly earlier sleep/wake times

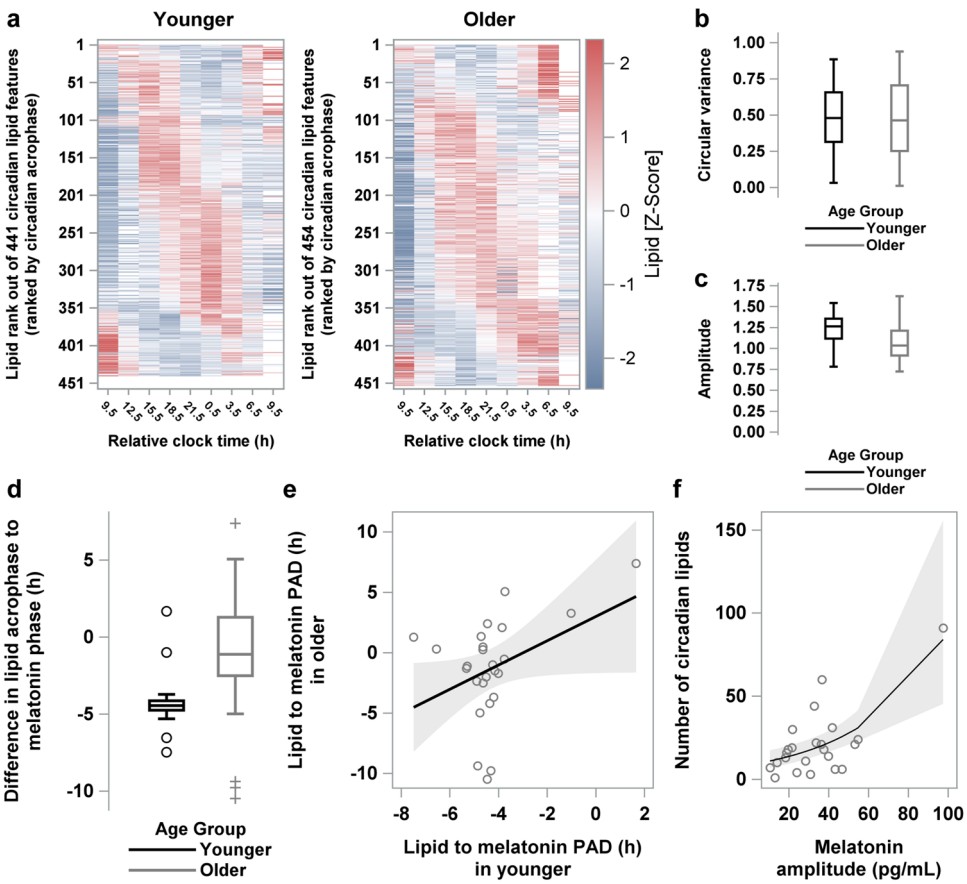

**Fig. 6 Comparative analysis of the timing and amplitude of the lipid species that were circadian in younger- or older-age individuals. a** Heat maps are shown for Z-scored plasma concentration levels of each lipid that was identified as circadian in each individual in the younger- and older-age groups. Lipids are ranked based on the time of their acrophase estimated from the Cosinor regression applied to individual-level data for each lipid. The box and whisker plots for (**b**) circular variance, (**c**) amplitude estimates for circadian lipids and (**d**) the phase angle difference of each of the 25 lipids that were circadian in both the younger and older groups relative to the group-mean melatonin phase (DLMO$_{25\%}$). Correlation between lipid acrophases and melatonin onset (**e**). Association between melatonin amplitude and (**f**) the number of lipids that were circadian. Data are plotted to a relative clock time with wake time assigned a value of 8:00 am. The box and whisker plots show the median, 25th and 75th percentile (box limits), the 10th and 90th percentiles (whiskers), and outlier points. Polar plots of acrophase of each circadian lipid in the younger and older age groups. Results are from $n = 24$ biologically independent samples in the younger ($n = 12$) and older ($n = 12$) age groups.

of the older individuals compared to the younger individuals. Likewise, the earlier phase of lipids may also be associated with an earlier central circadian phase observed in older individuals compared to younger individuals. Our study does not, however, allow identifying the underlying time cue(s) entraining these rhythms; therefore, we cannot identify what factors within the biochemical or behavioral cycle is contributing to this earlier shift in lipid phases.

Consistent with some[35–37] but not all[6,8] previous reports, no age-related differences between the phase angle of entrainment between the melatonin rhythm and wake time were observed. It is possible that the lack of difference between the age groups in phase angle between the melatonin rhythm and sleep was due to inadequate statistical power and the extensively screened healthy status of the participants. Notably, however, we found evidence of age-related differences in the phase angle of entrainment between melatonin phase, a marker of central clock output, and the average lipid acrophase, providing preliminary support for aging-related uncoupling of the central clock and peripheral outputs as contributors to age-associated metabolic dysfunction, even in relatively healthy individuals. Linear regression analyses showed a slope of 1.00, however, suggesting that the system-wide shift to an earlier phase in the older age group occurs in a coordinated

manner between physiologic systems. This result, however, should be interpreted cautiously. Firstly, the trend was not statistically significant ($p = 0.058$), secondly only a subset of lipids could be included in the analysis, and finally the results could not be verified at the individual level. Additional studies are therefore required to confirm and to understand this relationship.

Overall, the findings from our study suggest that there is an age-associated weakening of circadian regulation of lipid metabolism, which may contribute to the higher prevalence of metabolic dysfunction during aging. In mechanistic experiments, impaired circadian regulation of lipid metabolism has been shown to result from the loss of clock gene function [reviewed in refs. [21,38]]. Consistent with this evidence, clinical research has shown that misalignment of meals relative to the endogenous circadian phase adversely affects metabolic outcomes[39–41]. For example, postprandial glucose, insulin, and triglyceride responses are elevated during night time eating, as compared to daytime eating, both under simulated shiftwork and field-based shiftwork studies[42,43]. Therefore, potential changes in circadian phase of lipid metabolism, similar to changes in circadian phase of core body temperature, melatonin, and sleep/wake rhythms with aging, may contribute to metabolic dysregulation due to meals being misaligned relative to endogenous circadian rhythms in

older adults. Additionally, a reduction in amplitude of circadian lipid metabolism, potentially reflecting reduced strength in circadian regulation, may indicate greater susceptibility of lipid metabolism to perturbations induced by changes in environmental and behavioral factors such as sleep loss, shiftwork, or irregular rest/activity and feeding/fasting cycles. At the level of our experiment, we note that, during the CR protocol, small equicaloric snacks are taken at all phases of the circadian cycle, and an age-related reduction in the ability to properly metabolize such mistimed snacks may contribute to the effects that we observe. At the level of clinical relevance, we note that weakened circadian regulation might be both a direct and a somewhat orthogonal risk factor in the context of the way metabolic risk is conceptualized: the phrase somewhat orthogonal implies that circadian disruption-associated metabolic dysfunction might not require the normal factors postulated to cause metabolic disease, e.g., obesity, high fat/high sugar diets, genetic risk factors, low exercise, etc., while direct implies that people eating off-cycle have been shown to over-indulge in unhealthy dietary choices. Notably, by using the CR protocol, we have distributed any direct effects evenly across the circadian cycle in both age groups; therefore, direct effects are unlikely to be driving the observed effects.

The results from our study are generally in agreement with the only other prior study that assessed circadian regulation of the human plasma lipidome under CR[14]. The previous study found ~13% of 236 identified lipid species in young adults exhibited circadian variation at the group level, and between 5 to 33% at the individual level. We found ~25% of the identified lipid species to have circadian rhythmicity at the group level and between 3 to 58% at the individual level[14]. The difference in prevalence estimates may be due to the differences in protocol and study population. For example, the blood sampling frequency during the CR protocol was higher in our study (every 3 h) than in the previous study (every 4 h), which may have enabled us to detect more rhythms. Additionally, we used a nonlinear regression model that combined sinusoidal and linear terms, which was different from what was used in the previous study [Jonckheere-Terpstra-Kendall (JTK_CYCLE)][44], which included only a single sinusoidal term. Furthermore, our study population included a wide range of ages (~20–69 years) and female participants, whereas the previous study only included young men (21–28 years). Restricting our results to only the younger males and lipids that were circadian and without a linear component, we found that the number of lipids that exhibited circadian variation ranged between 3.3% to 32.8%, which is consistent with the previous finding. This, however, underscores the importance of further systematic testing of age and potentially sex differences in the circadian regulation of plasma lipids.

Our study has limitations. The current analysis combined data from two separate studies and although the combined number of participants between the two studies is comparable to sample sizes in previous studies examining circadian regulation of physiological outcomes including omics-based endpoints, the statistical power of the analyses could be improved with additional participants and more frequent sampling. Additionally, the larger sample size would have facilitated more extensive individual-level analyses. Moreover, although the functionally equivalent designs of the two independent studies enabled us to compare between them, studying both age groups at the same time with dedicated sample processing for lipidomics would improve sample quality and increase the potential lipid comparisons. Additionally, a longer CR would have enabled excluding the first few hours of data at the start of the CR, which may have removed any residual evoked masking effects from the previous sleep episode immediately prior to the start of the CR; however due to the relatively short 27-h length of the CR, such exclusion was not possible. A longer CR would have also enabled further assessment of the time-course profiles (circadian and non-circadian effects) over multiple circadian cycles to determine the resilience over time in circadian regulation. Furthermore, there were marginal differences in ambient conditions between the two studies; for example, light levels were higher in the older cohort [0.0087 W/m² (~3.3 lux)] than in the younger cohort [~0.001 W/m² (~0.5 lux)]. Both these levels, however, were significantly lower than the threshold of inducing circadian changes including circadian entrainment in humans and can be considered biologically inert[45,46]; therefore, no confounding is expected from these marginal differences. While circadian rhythms are typically modeled as sinusoidal profiles, as was done in the current study, it is possible that non-sinusoidal near-24 h periodic rhythms were undetected using our approach, resulting in a conservative estimate of the prevalence of circadian rhythms in the two age groups. Additionally, constant wakefulness and frequent meal consumption during the CR may have evoked responses that masked the underlying sinusoidal circadian profile, which may not have been captured using the linear term in our sinusoidal regression model. Future studies using higher sampling frequencies and alternative approaches to model periodic oscillations, or comparison to other meal schedules (i.e., 3 main meals unevenly spaced such as those used under the baseline study days) are required to enhance our current understanding of circadian regulation of the plasma lipidome. Another limitation of the study is the extensively screened healthy population. Both of these reduce our ability to generalize our findings. The circadian regulation of plasma lipids is likely different between healthy and clinical populations. Whether the changes in phase and amplitude observed with aging translate to clinical outcomes needs further investigation. Interestingly, the positive correlation between the amplitude of the melatonin peak and the number of lipid profiles that are circadian observed in our study, warrants further investigation to determine a causal relationship. It may also potentially provide a mechanism to improve endogenous circadian regulation of the human plasma lipidome in older individuals. Our study also does not allow identification of changes in mechanistic pathways that may explain the observed results. For example, glucocorticoids may play a role in entraining peripheral circadian rhythms[47–49], and investigating the mechanistic role of cortisol response in mediating the findings of the current study is warranted, specifically examining whether HPA activity is altered with healthy aging, as recently suggested (e.g.[50]) and in turn its effects on circadian regulation of peripheral circadian rhythms.

The work presented here, when placed in the context of decades of largely independent studies on aging and on circadian biology and the more recent investigations of the effects of aging on circadian regulation, suggests that a systematic investigation of age-related shifts—or lack thereof—in the broad linkages between lipids and the circadian system may be rich in several, complementary ways. Such investigations will help us to understand the impact of aging and age-related diseases on lipid metabolism, specifically whether the age-related changes, including changes in phase and amplitude, typically observed with markers of the central pacemaker are also observed with circadian rhythmicity of peripheral circadian metabolic processes such as lipid regulation. Conversely, lipids may serve as a test system for the regulation of the peripheral effects of circadian clocks in aging, as a readout for the effects of aging and age-associated disease on circadian biology, and as a potential causal factor linking aging to metabolic dysfunction. At the level of the timing of circadian phase, such investigations would answer whether the temporal shifts to earlier circadian phases observed in the centrally-controlled rhythms are paralleled by shifts in the peripheral clocks and their

associated physiologic rhythms, or whether this shift become a source of misalignment between central and peripheral rhythms that may contribute to age-associated metabolic dysfunction. At the level of the strength of circadian phase, such investigations would answer whether the reduction in amplitude observed in centrally-controlled rhythms extends to this aspect of the peripheral clocks and their associated physiologic rhythms, or whether this reduction becomes a weak point for circadian control of metabolism.

Overall, our study confirms robust circadian regulation of the human plasma lipidome in both younger and middle-aged individuals. There are significant changes in this regulation with healthy aging, including an advancement in circadian phase of the lipidome, reduced amplitude in the daily circadian variation of the lipidome and the lipid species that are under circadian regulation. Future studies are needed to explore the functional and clinical implications of these changes associated with healthy aging and determine whether additional changes are present in more typical older individuals.

## Methods

**Overview**. Semi-targeted lipidomics analysis was conducted on blood samples collected during CR protocols in two separate studies[22,23], both designed to assess the phase resetting effects of nocturnal light exposure in humans. The methodological details of each study and the primary outcome measures have been reported previously[22,23]. The methodological details relevant to the current study are reported below.

**Participants**. We used data from two studies conducted in the same facility with identical procedures relevant to this work, including data from 24 healthy individuals (9 females; mean ± SD age: 40.9 ± 18.2 years. The study by Scheuermaier et al.[22], included 12 healthy older participants [5 females, mean ( ± SD) age: 58.3 ± 4.2 years)] and the study by Rahman et al.[23], included 26 healthy younger participants (15 females; mean: 23.6 ± 3.3 years). The current report includes data from all participants from the older group, but only 12 participants (4 females, mean age: 23.5 ± 3.9 years) from the younger group. These 12 participants were selected at random from the cohort of 26 participants. All participants were studied in the Intensive Physiological Monitoring Unit in the Center for Clinical Investigation (CCI) at the Brigham and Women's Hospital, Boston, MA. All participants were determined to be healthy by physical, psychological, and ophthalmologic clinical exams.

For at least 3 weeks prior to admission, participants maintained a self-selected, consistent 8-h sleep-wake/light-dark schedule that was confirmed with calls to a time- and date-stamped voicemail and, for at least 7 days prior to beginning the inpatient phase, objective monitoring with actigraphy and photometry (Actiwatch-L, Minimitter Inc.). Throughout screening, participants were asked to refrain from the use of any prescription or nonprescription medications, supplements, recreational drugs, caffeine, alcohol, or nicotine. Compliance was verified by urine and blood toxicology during screening and urine toxicology upon entry to the unit. The study was reviewed and approved by the Partners Human Research Committee (Boston, Massachusetts, USA), and participants provided written informed consent prior to study. All relevant ethical regulations were followed in the conduct of this study.

**Protocol**. Each participant in the older cohort completed an 8-day inpatient study[22] whereas the participants in the younger cohort completed a 9-day inpatient protocol[23] (Supplementary Fig. 1). Both protocols began with 3 baseline days with 8-h sleep episodes every 24 h at each individual's self-selected bedtime maintained prior to the inpatient study. The baseline days were immediately followed by a CR protocol lasting ~27 h in the older cohort and ~50 h in the younger cohort. Following completion of the CR in both protocols, participants had an 8-h recovery sleep opportunity, a nocturnal light exposure intervention followed by a 30-h CR and a final 8-h recovery sleep and discharge. Lipidomics analysis was conducted only on samples from the initial CR and the data analysis was truncated for the younger cohort at 27 h to align with the duration of the CR in the older cohort.

**Constant routine**. Throughout the CR[20], participants were asked to remain awake while sitting in bed in a semirecumbent posture, and were continuously monitored by a trained staff member. Daily nutritional and fluid intake was divided into hourly isocaloric portions, with controlled nutrient intake (150 mEq Na + /100 mEq K+ [± 20%]). The isocaloric snacks were calculated using the Harris-Benedict equation calculating the 24-h need for sedentary behavior [basal energy expenditure × 1.3] diet, 2,500 mL fluids/24 h. Room temperature was maintained at ~23 ± 2 °C. Ambient light levels were set to ~0.001 W/m2 (~0.5 lux) at 137 cm from

the floor in the vertical plane for the younger cohort, and ~0.0087 W/m2 (~3.3 lux) at 137 cm from the floor in the vertical plane for the older cohort.

**Lipidomics**. Lipid extraction and LC-MS analysis were as previously described[51–54]. Briefly, lipids were extracted from 30 μL serum aliquots (whole blood collected in sodium heparin green top tubes) according to the method of Bligh and Dyer[55], substituting DCM for chloroform[51,56].

**LC-MS analysis**. LC separation of the lipid extracts was as previously described[51–53]. Briefly, LC analysis was performed on an Accela Quarternary HPLC pump (Thermo Scientific, San Jose, CA). Lipids were separated by gradient elution on an Accucore C18, 2.1 ×150 mm, 2.6 μm column (Thermo Scientific, Waltham). The column temperature was set to 55 °C. A flow rate of 260 μl/min was used and 10 μl of each sample was injected on column. Mobile phase A was comprised of 60:40 Acetonitrile:Water and mobile phase B was comprised of 90:10 Isopropanol:Acetonitrile. The mobile phases (both A and B) were modified with 5 mM ammonium formate + 0.1 % formic acid and 1 mM ammonium acetate + 0.1 % acetic acid in the positive and negative ionization modes, respectively.

MS Analysis was performed on an Exactive Benchtop Orbitrap Mass Spectrometer (Themo Fisher, San Jose, CA) equipped with a heated electrospray ionization (HESI-II) probe. The HESI-II probe was run on the ESI mode (i.e., the HESI vaporizer temperature was off). In both the positive and the negative ionization modes, the heated capillary (i.e., ion transfer tube) temperature was maintained at 280 °C, the sheath gas flow was set to 30 units and the auxiliary gas was set to 20 units. The spray voltage was set to 3.8 kV in the negative mode and 4 kV in the positive mode. The MS was tuned with 2.5 μg/mL PG (17:0/17:0) in the positive ionization mode[51–53]. The MS was also tuned with 2.5 μg/ml LPC (19:0) in the negative ionization mode[54]. Other pooled human plasma controls and purified biochemical standards were as previously reported[54].

Feature selection (framing) and relative quantitation (based on peak area) was done using SIEVE v2.1 (Thermo Fisher Scientific and Vast Scientific, Cambridge, MA). The framing parameters were as follows: 5 ppm for the m/z window, 1.00 min for the RT window, masses between m/z 120 and 2000. The intensity threshold was set at 1000, with 10,000 frames selected. A plasma sample analyzed in the middle of the analysis was used as a reference. Lipids were identified (C:DB) from an in-house built database using a pipeline written to match the features to the database (currently 746 lipids) and all putatively identified lipids were then manually curated.

Raw Data—older group, for ESI-NEG, 75% of lipids have CVs <28.3% with a median of 11.3%. For ESI-POS, 75% of lipids have CVs <28.8% with a median of 23.5%. Raw Data—younger group, for ESI-NEG, 75% of lipids have CVs <33.5% with a median of 24%. For ESI-POS, 75% of lipids have CVs <43.6% with a median of 39.7%. The above reported CV values are of the lipids measured in the pools that are run periodically to assess analytical variability. This was 174 lipids in the younger and 188 lipids in the older cohort.

Samples had been stored at least part time at −20 °C and showed some evidence of oxidation and/or degradation, but within subject, within group, and within study analyses all suggested the samples were of sufficient quality for analysis.

**Central circadian phase assessment**. Melatonin concentration in the younger individuals was determined by double-antibody radioimmunoassay with the Kennaway G280 antiserum (Specialty Assay Research Core Laboratory, Brigham and Women's Hospital). The plasma melatonin intra-assay coefficient of variation (%CV) was 8.5% at 5.9 pg/ml and 8.0% at 25.5 pg/ml, and the interassay %CV was 10.8% at 6.0 pg/ml and 14.5% at 23.1 pg/ml. Melatonin concentration in the older individuals was determined using the Bühlmann Melatonin Direct RIA kit (Bühlmann Laboratories, Schönenbuch, Switzerland) and the assays conducted by Solidphase, Inc., (Portland, ME). The plasma melatonin intra-assay coefficient of variation (%CV) was 6.7% at 4.6 pg/ml and 7.5% at 14.3 pg/ml, and the interassay %CV was 16.2% at 2.3 pg/ml and 7.2% at 18.7 pg/ml. The dim light melatonin onset time (DLMO$_{25\%}$) was used as the endogenous circadian phase marker and calculated as the time at which levels of melatonin rose above the 25% peak-to-trough amplitude threshold of a 3-harmonic waveform fitted to the first melatonin secretory episode during CR1.

**Statistics and reproducibility**. Analysis was performed on group-mean and individual-level data. For all analyses, unadjusted lipid concentrations were transformed to Z-scores [Z = $(x − μ)/σ$] for each individual using all available data for that individual from the ~27-h CR[14]. Time-series data were binned in 3-h intervals and then averaged across individuals to calculate group-mean profiles. Individual and group-mean profiles were fit with a Cosinor regression model that included one 24-h fundamental sinusoidal and one linear component [$Y = A*cos((2π(x − φ))/24) + ((m*x)+b)$], where $x$ is time since wake in hours, $A$ is the amplitude of the sinusoid, $φ$ is the acrophase (phase of the sinusoid), $m$ is the slope of the linear term, $b$ is the vertical intercept of the linear term[57–59]. The $φ$ parameter was bounded between 0 and 24 h from wake, and initial values for $φ$, A, $m$ and $b$ were set to 12, 0, 0, and 0, respectively. The non-linear regression was carried out using ordinary least squares regression and was considered significant if the

amplitude was significantly different from 0 at an alpha threshold of 0.05[57,59]. Whenever a significant nadir was detected by the regression model then the acrophase time was calculated as the peak 12 h later[57,59]. All amplitude-based analyses were conducted on the absolute value of the amplitude. Differences in the number of circadian lipids between the age groups were compared using Fisher's exact or $\chi^2$ tests, as appropriate. When comparing amplitudes, data were checked for normality and compared using generalized linear models (GLM) with a main effect of age groups, Kruskal-Wallis, or the Friedman test, as appropriate. Mean acrophases were calculated using circular statistics and the variance was calculated as the circular variance. Acrophases were compared between the age groups using the Watson and Williams test[24]. Circular variance was compared between age groups using generalized linear models or Kruskal-Wallis, as appropriate based on the distribution of the data. Correlation between acrophases and wake time or central circadian phase ($DLMO_{25\%}$) was assessed using circular correlation, and association between the number of circadian lipid profiles and melatonin amplitude was assessed using general estimating equations (GEE) with a Poisson distribution adjusted for overdispersion. All analyses were performed using SAS 9.4 (SAS Institute Inc., Cary, NC, USA). Circular statistics was performed using SAS macros developed by Mathias Kölliker[60,61].

**Reporting summary**. Further information on research design is available in the Nature Portfolio Reporting Summary linked to this article.

## Data availability

Individual-level lipidomics data with timing information are not publicly available due to containing information that could compromise the privacy of research participants. These data are available on request from the corresponding author, (BSK) or lead author (SAR) after necessary institutional agreements are established. Source data for all main figures are provided as Supplementary Data.

## Code availability

All statistical analyses were performed using SAS 9.4 (SAS Institute Inc., Cary, NC, USA) using standard SAS coding. Circular statistics was performed using SAS macros developed by Mathias Kölliker[60,61].

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

## Acknowledgements

The authors wish to thank the study participants; the technical, nursing, and dietary staff of the Brigham and Women's Center for Clinical Investigation; and the technical staff of the Division of Sleep and Circadian Disorders Sleep Core and Chronobiology Core for their assistance with data collection and participant monitoring. The work was supported by grants from the NIH: R01-HL132556 (BSK), R01-HL140335 (BSK), R01-HL114088 (EBK), R01-AG06072 (JFD), and R01-HL159207 (SAR). KS was supported by a T32 HL07901 and a NIA F32 AG316902. EBK was supported by NIH R01NS099055, U01NS114001, U54AG062322, R21DA052861, R21DA052861, R01NS114526-02S1, R01-HD071064, DoD W81XWH201076; and Leducq Foundation for Cardiovascular Research. The clinical research projects described were supported by NIH grant 1UL1 TR001102-01, 8UL1TR000170-05, UL1 RR025758, Harvard Clinical and Translational Science Center, from the National Center for Advancing Translational Science. The content is solely the responsibility of the authors and does not necessarily represent the official views of the National Center for Research Resources, the National Center for Advancing Translational Science or the National Institutes of Health.

## Author contributions

S.A.R., M.A.S.H., K.S., C.A.C., S.W.L., E.B.K., J.F.D. and B.S.K. were involved in the conception and design of the studies. B.S.K. initiated the research project. S.A.R., R.M.G., V.R.M., M.A.S.H., K.S., M.B., J.S.S. and B.S.K. collected and/or analyzed the data. All authors reviewed and approved the manuscript.

## Competing interests

R.M.G., V.R.M., M.B.B., J.S.S., J.F.D., and K.S. declare no competing interests. SAR holds patents for (1) Prevention of circadian rhythm disruption by using optical filters and (2) Improving sleep performance in subject exposed to light at night; SAR owns equity in Melcort Inc.; has provided paid consulting services to Sultan & Knight Limited, Bambu Vault LLC, Lucidity Lighting Inc.; and has received honoraria as an invited speaker and travel funds from Starry Skies Lake Superior, University of Minnesota Medical School, PennWell Corp., and Seoul Semiconductor Co. Ltd. These interests were reviewed and managed by Brigham and Women's Hospital and Partners HealthCare in accordance with their conflict of interest policies. MSH has provided limited consulting to The MathWorks, Inc. CAC reports grants and contracts to BWH from Dayzz Live Well, Delta Airlines, Jazz Pharma, Puget Sound Pilots, Regeneron Pharmaceuticals/Sanofi; is/was paid consultant/speaker for Inselspital Bern, Institute of Digital Media and Child Development, Klarman Family Foundation, M. Davis and Co, National Council for Mental Wellbeing, National Sleep Foundation, Physician's Seal, SRS Foundation, State of Washington Board of Pilotage Commissioners, Tencent, Teva Pharma Australia, With Deep, and Vanda Pharmaceuticals, in which CAC holds an equity interest; received travel support from Aspen Brain Institute, Bloomage International Investment Group, Inc., Dr. Stanley Ho Medical Development Foundation, German National Academy of Sciences, Ludwig-Maximilians-Universität München, National Highway Transportation Safety Administration, National Safety Council, National Sleep Foundation, Salk Institute for Biological Studies/Fondation Ipsen, Society for Research on Biological Rhythms, Stanford Medical School Alumni Association, Tencent Holdings, Ltd, and Vanda Pharmaceuticals; receives research/education gifts through BWH from Arbor Pharmaceuticals, Avadel Pharmaceuticals, Bryte, Alexandra Drane, Cephalon, DR Capital Ltd, Eisai, Harmony Biosciences, Jazz Pharmaceuticals, Johnson & Johnson, Mary Ann & Stanley Snider via Combined Jewish Philanthropies, NeuroCare, Inc., Optum, Philips Respironics, Regeneron, Regional Home Care, ResMed, Resnick Foundation (The Wonderful Company), San Francisco Bar Pilots, Sanofi SA, Schneider, Simmons, Sleep Cycle AB. Sleep Number, Sysco, Teva Pharmaceuticals, Vanda Pharmaceuticals; is/was an expert witness in legal cases, including those involving Advanced Power Technologies, Aegis Chemical Solutions, Amtrak; Casper Sleep Inc, C&J Energy Services, Catapult Energy Services Group, Covenant Testing Technologies, Dallas Police Association, Enterprise Rent-A-Car, Espinal Trucking/Eagle Transport Group/Steel Warehouse Inc, FedEx, Greyhound, Pomerado Hospital/Palomar Health District, PAR Electrical Contractors, Product & Logistics Services LLC/Schlumberger Technology, Puckett EMS, Puget Sound Pilots, Union Pacific Railroad, UPS, and Vanda Pharmaceuticals; serves as the incumbent of an endowed professorship given to Harvard by Cephalon; and receives royalties from McGraw Hill and Philips Respironics for the Actiwatch-2 and Actiwatch Spectrum devices. CAC's interests were reviewed and are managed by the Brigham and Women's Hospital and Mass General Brigham in accordance with their conflict-of-interest policies. SWL has had a number of commercial interests in the last 2 years (2019–21). His interests were reviewed and managed by Mass General Brigham in accordance with their conflict of interest policies. No interests are directly related to the research or topic reported in this paper but, in the interests of full disclosure, are outlined below. SWL has received consulting fees from the EyeJust Inc., Rec Room, Six Senses, and Stantec; and has current consulting contracts with Akili Interactive; Apex 2100 Ltd.; Consumer Sleep Solutions; Hintsa Performance AG; KBR Wyle Services, Light Cognitive; Lighting Science Group Corporation/HealthE; Mental Workout/Timeshifter, Sleep Standards and View Inc. He has received honoraria from Bloxhub/Lys, Danish Centre for Lighting, MIT, Roxbury Latin School, University of Toronto and Wiley; and royalties from Oxford University Press. He holds equity in iSleep Pty. He has received an unrestricted equipment gift from F. Lux Software LLC, and holds an investigator-initiated grant from F. Lux Software LLC. He has a Clinical Research Support Agreement and a Clinical Trials Agreement with Vanda Pharmaceuticals Inc. He is an unpaid Board Member of the Midwest Lighting Institute (non-profit). He was a Program Leader for the CRC for Alertness, Safety and Productivity, Australia, through an adjunct professor position at Monash University (2015–2019). He is currently a part-time adjunct faculty member at the University of Surrey. He has served as a paid expert in legal proceedings related to light, sleep and health. EBK reports consulting income from the American Academy of Sleep Medicine Foundation, Circadian Therapeutics, National Sleep Foundation, Sleep Research Society Foundation, and Yale University Press; has received travel support from the European Biological Rhythms Society and EPFL Pavilion; and is an unpaid member of the scientific advisory board of Chronsulting. Klerman's partner is founder, chief scientific officer of Chronsulting. BSK is the inventor on general metabolomics-related IP that has been licensed to Metabolon via Weill Medical College of Cornell University and for which he received royalty payments via Weill Medical College of Cornell University. He also consulted for and has an equity interest in the company. Metabolon offers biochemical profiling services and is developing molecular diagnostic assays detecting and monitoring disease. Metabolon has no rights or proprietary access to the research results presented and/or new IP generated under these grants/studies. BSK's interests were reviewed by the Brigham and Women's Hospital and Mass General Brigham (as Partners Healthcare) in accordance with their institutional policy. Accordingly, upon review, the institution determined that BSK's financial interest in Metabolon does not create a significant financial conflict of interest (FCOI) with this research. The addition of this statement where appropriate was explicitly requested and approved by BWH.
