## [Peer Review File · Communications Biology]

Reviewers' comments:

Reviewer #1 (Remarks to the Author):

Shadab A. Rahman and colleagues show that endogenous circadian rhythmicity of the human plasma lipidome during CR is preserved during aging but comes with a significant reduction in amplitude, advanced acrophase shift, and an altered relationship between melatonin phase and lipid acrophases. The main limitation of this study is the use of samples from two separate small studies. However, the authors clearly describe this on multiple occasions in the manuscript.

I would like to compliment the authors on their comprehensive work and the way they were able to describe the data in a constructive manner. I have no further comments and therefore would like to endorse this manuscript for publication.

Reviewer #2 (Remarks to the Author):

There is a growing recognition of the link between circadian rhythm disruption and metabolic dysregulation and disease. This is an increasing concern within our society, particularly for the elderly. Therefore, studies exploring this link are of importance. In this study, Rahman and colleagues investigated the effects of healthy ageing on the circadian rhythms of human plasma lipidome. Their results support the main conclusion of their study, indicating that intrinsic circadian rhythm in the human plasma lipidome is preserved during healthy ageing, but its characteristics (amplitude and phase) are significantly altered.

The experimental design and analyses are appropriate, and, in most places, the results are well-reported. Overall, the manuscript is well and succinctly written and the message is relevant and timely. However, there are a few avenues that I would like the authors to consider for improvement.

- 1) The number of male and female participants – for completeness the authors should consider categorically stating the number of males and females used. For example, on Page 6: the sentence beginning “The goal was addressed.....(12 individuals per group, 4 and 5 females in the younger and older age groups, respectively).....”. The numerical calculation for the participants here is difficult to follow when first read. This also extends to the abstract and some sections in the methods.
 - 2) Why were the ambient lighting conditions different for young and older cohorts during the CR? Would this not confound the comparisons?
 - 3) Related to the above, the authors should consider stating clearly in the methods the LD cycle or constant conditions that the participants experienced at specific stages in the study.
 - 4) In the discussion, the authors focused on central clock mechanisms to explain some of the altered circadian rhythm characteristics in the plasma lipidome. Is there any study on the ageing peripheral clock activity that could also explain some of the disruptions seen?
 - 5) Related to the above, I find the discussion rather long, and perhaps the authors should consider streamlining it.
 - 6) Would it make sense to combine the Cosinor plots in Figures 2 and 3 to produce a single figure (in different colours)? Especially for panels A, B, and C (combining D may make the figure too busy).
- Minor
- 7) Figures: Some of the shades used to discriminate between groups are hard to see in the figures (for example figure 1). Could these be plotted in colour?
 - 8) In figure 2 legend “The 95% CI region of the regression....” define “CI” (which I assume is the confidence interval). Also, would be helpful for the reader if the authors write the lipid names in full at the end of this legend.
 - 9) Brief legends for tables will be very helpful, especially if the full definitions of acronyms used are included.
 - 10) Typographical errors: For example, Page 27: - last paragraph 3rd line.
 - 11) A few punctuation errors, such as on page 20, second paragraph, line 12.

Reviewer #3 (Remarks to the Author):

This is a very elegant and timely study characterizing peripheral circadian rhythmicity in the plasma lipidome as a marker of systemic metabolism, which potentially may be altered with aging. The strength of the article, in my opinion, is that the assessment is of healthy aging. Lipidomics analysis was performed on plasma samples 'collected in 12 healthy young [hereafter "younger", 4 females, mean±SD age: 23.5±3.9 years] and 12 healthy middle-aged [hereafter "older", 5 females, 58.3±4.2 years] individuals every 3 h throughout a 27-h constant routine (CR) protocol, which allows separation of behaviorally-induced changes in physiologic outcomes from endogenously generated oscillations in them'. The only concerns are : low n number may contribute to low power for the statistics. The Analysis only includes measurements of melatonin, which seems very odd. As metabolism is objectively regulated by cortisol, and cortisol circadian rhythm is described as an important factor in the introduction of the paper, I find it very unusual that there are no Cortisol measurement of the subjects in the study. In particular, during the CR protocol, it will be important to know whether the different cohorts respond differently to the CR regime. For example, if the 'Older' group have a different cortisol response to the CR protocol that the 'younger' group' then this may affect their metabolic profiles. If blood samples (or even saliva samples) are available, then cortisol measurements should definitely be included in the analysis, and will hopefully aid in the interpretation of any differences between 'young' and 'older' groups

Minor comments: There were a many typographical errors in the manuscript. However without line numbering, it becomes a somewhat arduous task to give feedback on this. I am happy to do so in the next draft if line numbers are included.

Second minor comment : Titles and legends of figures should be more informative. Many abbreviations are used (and rightly so) but a reference point (even in a supplementary figure) might help readers in 'following' and interpretation of this data-rich article.

Reviewers' comments:

Reviewer #1 (Remarks to the Author):

1. I would like to compliment the authors on their comprehensive work and the way they were able to describe the data in a constructive manner. I have no further comments and therefore would like to endorse this manuscript for publication.

Response: We thank the reviewer for commendation of our work and endorsing the report for publication.

Reviewer #2 (Remarks to the Author):

1) The number of male and female participants – for completeness the authors should consider categorically stating the number of males and females used. For example, on Page 6: the sentence beginning “The goal was addressed.....(12 individuals per group, 4 and 5 females in the younger and older age groups, respectively).....”. The numerical calculation for the participants here is difficult to follow when first read. This also extends to the abstract and some sections in the methods.

Response: We have made the following edits to address the reviewer’s concern:

Abstract (Page 4, Lines 5-8): “Lipidomics analysis was carried out retrospectively on plasma samples collected in 24 healthy individuals (9 females; mean±SD age: 40.9±18.2 years) every 3 h throughout a 27-h constant routine (CR) protocol.”

Page 6 (Page 6, Lines 19-22): “The goal was addressed by comparing Cosinor-defined circadian profiles of lipid features between healthy young and older adults (24 individuals [9 females] in total; 12 individuals per group, 4 and 5 females in the younger and older age groups, respectively)...”

Methods (Page 25, Lines 11-13): “We used data from two studies conducted in the same facility with identical procedures relevant to this work, including data from 24 healthy individuals (9 females; mean±SD age: 40.9±18.2 years.”

2) Why were the ambient lighting conditions different for young and older cohorts during the CR? Would this not confound the comparisons?

Response: The lighting conditions were different between the studies as they were designed to be consistent with other studies that each were run in parallel with. Although the ambient light levels were higher in the older cohort [0.0087 W/m² (~3.3 lux)] than in the younger cohort [approximately 0.001 W/m² (~0.5 lux)], light levels for both the younger and older groups were significantly lower than the threshold of inducing circadian changes including circadian entrainment in humans and can be considered biologically inert (Gronfier et al., PNAS 2007; Zeitzer et al., 2004); therefore, we do not expect any confounding from the marginally different lighting levels. We have added the following section to the discussion to address this issue (Page 23, Lines 9-15): “Furthermore, there were marginal differences in ambient conditions between the two studies; for example, light levels were higher in the older cohort [0.0087 W/m² (~3.3 lux)] than in the younger cohort [approximately 0.001 W/m² (~0.5 lux)]. Both these levels, however, were significantly lower than the threshold of inducing circadian changes including circadian entrainment in humans and can be considered biologically inert (Gronfier et al., PNAS 2007; Zeitzer et al., J Physiol 2001); therefore, no confounding is expected from these marginal differences.”

3) Related to the above, the authors should consider stating clearly in the methods the LD cycle or constant conditions that the participants experienced at specific stages in the study.

Response: We have added light/dark (LD) cycle information to the protocol rasters that were presented in Supplemental Figure 1. The same rasters also show when study participants experienced constant conditions in each protocol. The constant conditions are described in the methods section (Page 26, Lines 21 to Page 27, Line 4).

Revised Figure and Legend:

Supplemental Figure 1. Study protocol for assessing circadian rhythms in the human plasma lipidome. All study events were timed according to each individual's schedule, which was maintained for at least three weeks prior to starting the inpatient part of the study. Example study raster for an individual with habitual self-selected sleep between midnight and 8:00 h in the (A) younger and (B) older age groups. Black bars represent scheduled sleep in darkness (time in bed), and white bars represent being awake with indoor intensity light [$\sim 0.23 \text{ W/m}^2$ ($\sim 88 \text{ lux}$)] when measured in the vertical plane at a height of 137 cm]. Gray bars represent being awake in dim light condition [older: $\sim 0.0087 \text{ W/m}^2$ ($\sim 3.3 \text{ lux}$); younger: $\sim 0.001 \text{ W/m}^2$ ($\sim 0.5 \text{ lux}$), when measured in the vertical plane at a height of 137 cm], and gray dotted bars represent constant routine intervals in dim lighting. Hashed bars represent the light exposure intervention. Only data from the first 27 hours of the first constant routine for both groups was included in the current study.

4) In the discussion, the authors focused on central clock mechanisms to explain some of the altered circadian rhythm characteristics in the plasma lipidome. Is there any study on the ageing peripheral clock activity that could also explain some of the disruptions seen?

Response: To our knowledge, this is the first demonstration of aging-related changes in endogenously regulated circadian peripheral rhythms in humans. A recent report (Talamanca et al., Science 2023) examining 24-hour rhythmicity in gene expression profiles from RNA-seq samples collected from 914 donors across 46 tissues found that rhythmicity was largely conserved across tissues in both younger (38 ± 9 years) and older (65 ± 3 years) groups, rhythmic gene expression was largely damped in older individuals. Importantly, however, these rhythms were not assessed under constant routine conditions, which precludes conclusively determining whether these observed changes are strictly due to changes in endogenous circadian regulation with aging or the reflect changes in the impact of external behavioral and environmental influencers (e.g., sleep/wake, activity, feeding) on circadian regulation and peripheral clocks in humans. Nonetheless, these results are largely consistent with our findings that endogenous circadian regulation of peripheral rhythms in lipids is conserved across aging but significantly damped (i.e., lower amplitude) in older humans. Furthermore, both of our results are consistent with previous reports from studies in animal models that show that aging affects peripheral clocks largely with phase advances and damped oscillations, which, data suggests, is due to reduced

coupling strength of the SCN to peripheral tissues (Yamazaki et al., PNAS 2002; Tahara et al., Aging 2017). We have included this discussion in the revised report (Page 19, Lines 9-24).

5) Related to the above, I find the discussion rather long, and perhaps the authors should consider streamlining it.

Response: We have revised the discussion section throughout to consolidate and reduce the overall length.

6) Would it make sense to combine the Cosinor plots in Figures 2 and 3 to produce a single figure (in different colours)? Especially for panels A, B, and C (combining D may make the figure too busy).

Response: We have edited both Figure 2 and 3 to move the exemplary cosinor plots into a combined new Supplemental Figure 2. We have chosen to keep the panels for the younger and older groups separate as overlaying the plots even as different colors makes it difficult to discern the individual plots and corresponding 95% CI bands.

Revised Supplemental Figure 2:

Supplemental Figure 2. Example time-course profile of a lipid species best modelled with a Cosinor regression model in which the significant components were both the 24-h (sinusoidal) harmonic and the linear component (combined), only the sinusoidal component (circadian only), or only the linear component (linear only) in the younger (A, B, C) and older (D, E, F) groups, respectively. Group-mean estimates (\pm SE) of the Z-score transformed data in each 3-h time bin during the 27-h constant routine are shown (\bullet). Solid black line represents the lipid concentration predicted by the Cosinor regression. The 95% confidence interval (CI) region of the regression is shown as the gray shaded band.

7) Figures: Some of the shades used to discriminate between groups are hard to see in the figures (for example figure 1). Could these be plotted in colour?

Response: We have used color to primarily differentiate between lipid species or time points. We hope that the higher resolution images in the final version, if accepted, will help with discriminating groups plotted in black and grayscale.

8) In figure 2 legend "The 95% CI region of the regression...." define "CI" (which I assume is the confidence interval). Also, would be helpful for the reader if the authors write the lipid names in full at the end of this legend.

Response: We have revised Figure 2 and 3 that required reference to "CI", which is now relevant to Supplemental Figure 2. We have defined CI, confidence interval. We have also defined the abbreviated lipid subclass names in Figures 2, 3, 4 and 5.

9) *Brief legends for tables will be very helpful, especially if the full definitions of acronyms used are included.*

Response: We have added legends for both Table 1 and 2 to define abbreviations.

10) *Typological errors: For example, Page 27: - last paragraph 3rd line.*

Response: We apologize for these errors and corrected them throughout the report.

11) *A few punctuation errors, such as on page 20, second paragraph, line 12.*

Response: We apologize for these errors and corrected them throughout the report.

Reviewer #3 (Remarks to the Author):

1. *The only concerns are : low n number may contribute to low power for the statistics.*

Response: We agree with the reviewer that we were likely underpowered to observe statistical differences for some of the comparisons, and this limitation is acknowledged in the report (e.g., Page 22, Line 22). The current results, however, will be foundational for estimating sample size requirements for future studies adequately powered to detect differences in outcomes that were not statistically significantly different in the current study.

2. *The Analysis only includes measurements of melatonin, which seems very odd. As metabolism is objectively regulated by cortisol, and cortisol circadian rhythm is described as an important factor in the introduction of the paper, I find it very unusual that there are no Cortisol measurement of the subjects in the study. In particular, during the CR protocol, it will be important to know whether the different cohorts respond differently to the CR regime. For example, if the 'Older' group have a different cortisol response to the CR protocol that the 'younger' group' then this may affect their metabolic profiles. If blood samples (or even saliva samples) are available, then cortisol measurements should definitely be included in the analysis, and will hopefully aid in the interpretation of any differences between 'young' and 'older' groups*

Response: We thank the reviewer for raising this important point. Cortisol rhythms also exhibit reduction in amplitude and advanced phase with aging, consistent with the aging related changes in melatonin rhythm (Touitou Y, et al., *J Endocrinol* 1982; Van Cauter et al., *J Clin Endocrinol Metab* 1996; Sherman et al., *J Clin Endocrinol Metab* 1985). We have cortisol results in the younger but not the older group and the blood samples are no longer available from either cohort. Therefore, we cannot compare the available cortisol results between the age groups. We agree with the reviewer that investigating the mechanistic role of cortisol response is interesting, that is examining whether HPA activity is altered with healthy aging, as recently suggested (e.g., Stamou et al., *Maturitas* 2023) and in turn its effects on circadian regulation of peripheral rhythms; however, that is beyond the scope of the current work and will require future studies specifically designed to assess that relationship. We have addressed this as a limitation of the current study (Page 24, Lines 7-13) as follows:

“Our study also does not allow identification of changes in mechanistic pathways that may explain the observed results. For example, glucocorticoids may play a role in entraining peripheral circadian rhythms, and investigating the mechanistic role of cortisol response in mediating the findings of the current study is warranted, specifically examining whether HPA activity is altered with healthy aging, as

recently suggested (e.g., Stamou et al., Maturitas 2023) and in turn its effects on circadian regulation of peripheral circadian rhythms.”

3. Minor comments: There were a many typographical errors in the manuscript. However without line numbering, it becomes a somewhat arduous task to give feedback on this. I am happy to do so in the next draft if line numbers are included.

Response: We apologize for these errors and corrected them throughout the report. We have added line numbering throughout the text.

4. Second minor comment: Titles and legends of figures should be more informative. Many abbreviations are used (and rightly so) but a reference point (even in a supplementary figure) might help readers in 'following' and interpretation of this data-rich article.

Response: We have revised the end of the introduction section which precludes to the results section to outline the overall analytic strategy.

REVIEWERS' COMMENTS:

Reviewer #2 (Remarks to the Author):

I would like to thank the authors for their consideration and compliment them for their excellent work. I have no further comments and would very much like to endorse their manuscript for publication.

Reviewer #3 (Remarks to the Author):

The authors have satisfied all reviewers' concerns.

REVIEWERS' COMMENTS:

Reviewer #2 (Remarks to the Author):

I would like to thank the authors for their consideration and compliment them for their excellent work. I have no further comments and would very much like to endorse their manuscript for publication.

Response: We thank the reviewer for the helpful comments, commendation of our work and endorsing the report for publication.

Reviewer #3 (Remarks to the Author):

The authors have satisfied all reviewers' concerns.

Response: We thank the reviewer for the helpful comments, and endorsing the report for publication.